# The Multivariate Theory of Connections †

**Daniele Mortari ***  **and Carl Leake *** 

Aerospace Engineering, Texas A&M University, College Station, TX 77843, USA

* Correspondence: mortari@tamu.edu (D.M.); leakec@tamu.edu (C.L.); Tel.: +1-979-845-0734 (D.M.)

† This paper is an extended version of our paper published in Mortari, D. "The Theory of Connections: Connecting Functions." IAA-AAS-SciTech-072, Forum 2018, Peoples' Friendship University of Russia, Moscow, Russia, 13–15 November 2018.

**Abstract:** This paper extends the univariate Theory of Connections, introduced in (Mortari, 2017), to the multivariate case on rectangular domains with detailed attention to the bivariate case. In particular, it generalizes the bivariate Coons surface, introduced by (Coons, 1984), by providing analytical expressions, called *constrained expressions*, representing *all* possible surfaces with assigned boundary constraints in terms of functions and arbitrary-order derivatives. In two dimensions, these expressions, which contain a freely chosen function, $g(x, y)$, satisfy all constraints no matter what the $g(x, y)$ is. The boundary constraints considered in this article are Dirichlet, Neumann, and any combinations of them. Although the focus of this article is on two-dimensional spaces, the final section introduces the *Multivariate Theory of Connections*, validated by mathematical proof. This represents the multivariate extension of the Theory of Connections subject to arbitrary-order derivative constraints in rectangular domains. The main task of this paper is to provide an analytical procedure to obtain constrained expressions in any space that can be used to transform constrained problems into unconstrained problems. This theory is proposed mainly to better solve PDE and stochastic differential equations.

**Keywords:** interpolation; constraints; embedded constraints

## 1. Introduction

The Theory of Connections (ToC), as introduced in [1], consists of a general analytical framework to obtain *constrained expressions*, $f(x)$, in one-dimension. A constrained expression is a function expressed in terms of another function, $g(x)$, that is freely chosen and, no matter what the $g(x)$ is, the resulting expression always satisfies a set of $n$ constraints. ToC generalizes the one-dimensional interpolation problem subject to $n$ constraints using the general form,

$$f(x) = g(x) + \sum_{k=1}^{n} \eta_k \, p_k(x), \tag{1}$$

where $p_k(x)$ are $n$ user-selected linearly independent functions, $\eta_k$ are derived by imposing the $n$ constraints, and $g(x)$ is a *freely chosen* function subject to be *defined and nonsingular* where the constraints are specified. Besides this requirement, $g(x)$ can be any function, including, discontinuous functions, delta functions, and even functions that are undefined in some domains. Once the $\eta_k$ coefficients have been derived, then Equation (1) satisfies all the $n$ constraints, *no matter what the $g(x)$ function is*.

Constrained expressions in the form given in Equation (1) are provided for a wide class of constraints, including constraints on points and derivatives, linear combinations of constraints, as well as infinite and integral constraints [2]. In addition, weighted constraints [3] and point constraints

on continuous and discontinuous periodic functions with assigned period can also be obtained [1]. How to extend ToC to inequality and nonlinear constraints is currently a work in progress.

The Theory of Connections framework can be considered the generalization of interpolation; rather than providing a class of functions (e.g., monomials) satisfying a set of $n$ constraints, it derives *all* possible functions satisfying the $n$ constraints by spanning all possible $g(x)$ functions. This has been proved in Ref. [1]. A simple example of a constrained expression is,

$$f(x) = g(x) + \frac{x(2x_2 - x)}{2(x_2 - x_1)} \left[ \dot{y}_1 - \dot{g}(x_1) \right] + \frac{x(x - 2x_1)}{2(x_2 - x_1)} \left[ \dot{y}_2 - \dot{g}(x_2) \right]. \tag{2}$$

This equation always satisfies $\left. \dfrac{\mathrm{d}f}{\mathrm{d}x} \right|_{x_1} = \dot{y}_1$ and $\left. \dfrac{\mathrm{d}f}{\mathrm{d}x} \right|_{x_2} = \dot{y}_2$, as long as $\dot{g}(x_1)$ and $\dot{g}(x_2)$ are defined and nonsingular. In other words, *the constraints are embedded into the constrained expression.*

Constrained expressions can be used to transform constrained optimization problems into unconstrained optimization problems. Using this approach, fast least-squares solutions of linear [4] and nonlinear [5] ODE have been obtained at machine error accuracy and with low (actually, very low) condition number. Direct comparisons of ToC versus MATLAB's ode45 [6] and Chebfun [7] have been performed on a small test of ODE with excellent results [4,5]. In particular, the ToC approach to solve ODE consists of a unified framework to solve IVP, BVP, and multi-value problems. The extension of differential equations subject to component constraints [8] has opened the possibility for ToC to solve *in real-time* a class of direct optimal control problems [9], where the constraints connect state and costate.

This study first extends the Theory of Connections to two-dimensions by providing, for rectangular domains, *all* surfaces that are subject to: (1) Dirichlet constraints; (2) Neumann constraints; and (3) any combination of Dirichlet and Neumann constraints. This theory is then generalized to the Multivariate Theory of Connections which provide in $n$-dimensional space all possible manifolds that satisfy boundary constraints on the value and boundary constraints on any-order derivative.

This article is structured as follows. First, it shows that the one-dimensional ToC can be used in two dimensions when the constraints (functions or derivatives) are provided along one axis only. This is a particular case, where the original univariate theory [1] can be applied with basically no modifications. Then, a two dimensional ToC version is developed for Dirichlet type boundary constraints. This theory is then extended to include Neumann and mixed type boundary constraints. Finally, the theory is extended to $n$-dimensions and to incorporate arbitrary-order derivative boundary constraints followed by a mathematical proof validating it.

## 2. Manifold Constraints in One Axis, Only

Consider the function, $f(\boldsymbol{x})$, where $f : \mathbb{R}^n \to \mathbb{R}^1$, subject to one constraint manifold along the $i$th variable, $x_i$, that is, $f(\boldsymbol{x})|_{x_i=v} = c(\boldsymbol{x}_i^v)$. For instance, in 3-D space, this can be the surface constraint, $f(x, y, z)|_{y=\pi} = c(x, \pi, z)$. *All manifolds* satisfying this constraint can be expressed using the additive form provided in Ref. [1],

$$f(\boldsymbol{x}) = g(\boldsymbol{x}) + [c(\boldsymbol{x}_i^v) - g(\boldsymbol{x}_i^v)]$$

where $g(\boldsymbol{x})$ is a freely chosen function that must be defined and nonsingular at the constraint coordinates. When $m$ manifold constraints are defined along the $x_i$-axis, then the 1-D methodology [1] can be applied as it is. For instance, the constrained expression subject to $m$ constraints along the $x_i$ variable evaluated at $x_i = w_k$, where $k \in [1, m]$, that is, $f(\boldsymbol{x})|_{x_i=w_k} = c(\boldsymbol{x}_i^{w_k})$, is,

$$f(\boldsymbol{x}) = g(\boldsymbol{x}) + \sum_{k=1}^{m} \left\{ [c(\boldsymbol{x}_i^{w_k}) - g(\boldsymbol{x}_i^{w_k})] \prod_{j \neq k} \frac{x_i - w_j}{w_k - w_j} \right\}. \tag{3}$$

Note that this equation coincides with the Waring interpolation form (better known as Lagrangian interpolation form) [10] if the free function vanishes, $g(x) = 0$.

## 2.1. Example #1: Surface Subject to Four Function Constraints

The first example is designed to show how to use Equation (3) with mixed, continuous, discontinuous, and multiple constraints. Consider the following four constraints,

$$c(x, -2) = \sin(2x), \quad c(x, 0) = 3 \cos x \, [(x+1) \bmod(2)], \quad c(x, 1) = 9 \, e^{-x^2}, \quad \text{and} \quad c(x, 3) = 1 - x.$$

This example highlights that the constraints and free-function may be discontinuous by using the modular arithmetic function. The result is a surface that is continuous in $x$ at some coordinates (at $y = -2$, 1, and 3) and discontinuous at $y = 0$. The surfaces shown in Figures 1 and 2 were obtained using two distinct expressions for the free function, $g(x, y)$.

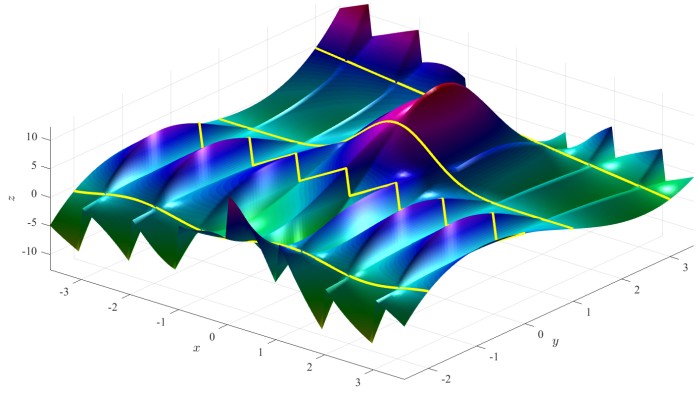

**Figure 1.** Surface obtained using function $g(x, y) = 0$ (simplest surface).

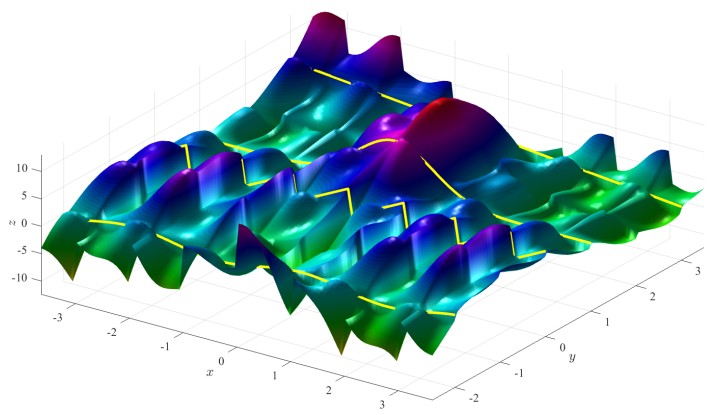

**Figure 2.** Surface obtained using function $g(x, y) = x^2 \, y - \sin(5x) \cos(4 \bmod(y, 1))$.

## 2.2. Example #2: Surface Subject to Two Functions and One Derivative Constraint

This second example is provided to show how to use the general approach given in Equation (1) and described in [1], when derivative constraints are involved. Consider the following three constraints,

$$c(x, -2) = \sin(2x), \qquad c_y(x, 0) = 0, \qquad \text{and} \qquad c(x, 1) = 9 \, e^{-x^2}.$$

Using the functions $p_1(y) = 1$, $p_2(y) = y$, and $p_3(y) = y^2$, the constrained expression form satisfying these three constraints assumes the form,

$$f(x, y) = g(x, y) + \eta_1(x) + \eta_2(x) \, y + \eta_3(x) \, y^2. \tag{4}$$

The three constraints imply the constraints,

$$\begin{aligned}
\sin(2x) &= g(x,-2) + \eta_1 - 2\eta_2 + 4\eta_3 \\
0 &= g_y(x,0) + \eta_2 \\
9\,e^{-x^2} &= g(x,1) + \eta_1 + \eta_2 + \eta_3,
\end{aligned}$$

from which the values of the $\eta_k$ coefficients,

$$\begin{aligned}
\eta_1 &= 2g_y(x,0) + 12\,e^{-x^2} - \frac{\sin(2x)}{3} + \frac{1}{3}g(x,-2) - \frac{4}{3}g(x,1) \\
\eta_2 &= -g_y(x,0) \\
\eta_3 &= \frac{\sin(2x)}{3} - \frac{1}{3}g(x,-2) - g_y(x,0) - 3\,e^{-x^2} + \frac{1}{3}g(x,1),
\end{aligned}$$

can be derived. After substituting these coefficients into Equation (4), the constrained expression that always satisfies the three initial constraints is obtained. Using this expression and two different free functions, $g(x,y)$, we obtained the surfaces shown in Figures 3 and 4, respectively. The constraint $c_y(x,0) = 0$, difficult to see in both figures, can be verified analytically.

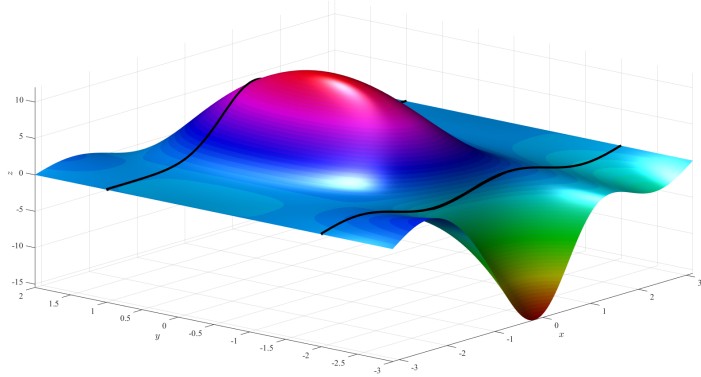

**Figure 3.** Surface obtained using function $g(x,y) = 0$ (simplest surface).

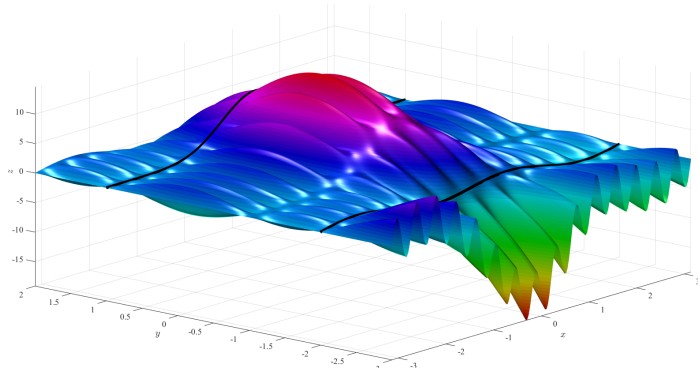

**Figure 4.** Surface obtained using function $g(x,y) = 3x^2y - 2\sin(15x)\cos(2y)$.

## 3. Connecting Functions in Two Directions

In this section, the Theory of Connections is extended to the two-dimensional case. Note that dealing with constraints in two (or more) directions (functions or derivatives) requires particular attention. In fact, two orthogonal constraint functions cannot be completely distinct as they intersect at one point where they need to match in value. In addition, if the formalism derived for the 1-D case is applied to 2-D case, some complications arise. These complications are highlighted in the following simple clarifying example.

Consider the two boundary constraint functions, $f(x,0) = q(x)$ and $f(0,y) = h(y)$. Searching the constrained expression as originally done for the one-dimensional case implies the expression,

$$f(x,y) = g(x,y) + \eta_1\, p_1(x,y) + \eta_2\, p_2(x,y).$$

The constraints imply the two constraints,

$$\begin{cases} q(x) = g(x,0) + \eta_1\, p_1(x,0) + \eta_2\, p_2(x,0) \\ h(y) = g(0,y) + \eta_1\, p_1(0,y) + \eta_2\, p_2(0,y). \end{cases}$$

To obtain the values of $\eta_1$ and $\eta_2$, the determinant of the matrix to invert is $p_1(x,0)\, p_2(0,y) - p_1(0,y)\, p_2(x,0)$. This determinant is $y$ by selecting $p_1(x,y) = 1$ and $p_2(x,y) = y$, or it is $x$ by selecting $p_1(x,y) = x$ and $p_2(x,y) = 1$. Therefore, to avoid singularities, this approach requires paying particular attention to the domain definition and/or on the user-selected functions, $p_k(x,y)$. To avoid dealing with these issues, a new (equivalent) formalism to derive constrained expressions is devised for the higher dimensional case.

The Theory of Connections extension to the higher dimensional case (with constraints on all axes) can be obtained by re-writing the constrained expression into an equivalent form, highlighting a general and interesting property. Let us show this by an example. Equation (2) can be re-written as,

$$f(x) = \underbrace{\frac{x(2x_2 - x)}{2(x_2 - x_1)}\dot{y}_1 + \frac{x(x - 2x_1)}{2(x_2 - x_1)}\dot{y}_2}_{A(x)} + \underbrace{g(x) - \frac{x(2x_2 - x)}{2(x_2 - x_1)}\dot{g}_1 - \frac{x(x - 2x_1)}{2(x_2 - x_1)}\dot{g}_2}_{B(x)}. \tag{5}$$

These two components, $A(x)$ and $B(x)$, of a constrained expression have a specific general meaning. The term, $A(x)$, represents an (*any*) interpolating function satisfying the constraints while the $B(x)$ term represents *all* interpolating functions that are vanishing at the constraints. Therefore, the generation of all functions satisfying multiple orthogonal constraints in $n$-dimensional space can always be expressed by the general form, $f(x) = A(x) + B(x)$, where $A(x)$ is *any* function satisfying the constraints and $B(x)$ must represent *all* functions vanishing at the constraints. Equation $f(x) = A(x) + B(x)$ is actually an alternative general form to write a *constrained expression*, that is, an alternative way to generalize interpolation: rather than derive a class of functions (e.g., monomials) satisfying a set of constraints, it represents *all* possible functions satisfying the set of constraints.

To prove that this additive formalism can describe *all* possible functions satisfying the constraints is immediate. Let $f(x)$ be all functions satisfying the constraints and $y(x) = A(x) + B(x)$ be the sum of a specific function satisfying the constraints, $A(x)$, and a function, $B(x)$, representing all functions that are null at the constraints. Then, $y(x)$ will be equal to $f(x)$ *iff* $B(x) = f(x) - A(x)$, representing all functions that are null at the constraints.

As shown in Equation (5), once the $A(x)$ function is obtained, then the $B(x)$ function can be immediately derived. In fact, $B(x)$ can be obtained by subtracting the $A(x)$ function, where all the constraints are specified in terms of the $g(x)$ free function, from the free function $g(x)$. For this reason, let us write the general expression of a constrained expression as,

$$f(x) = A(x) + g(x) - A(g(x)), \tag{6}$$

where $A(g(x))$ indicates the function satisfying the constraints where the constraints are specified in term of $g(x)$.

The previous discussion serves to prove that the problem of extending Theory of Connections to higher dimensional spaces consists of the problem of finding the function, $A(x)$, only. In two

dimensions, the function $A(\boldsymbol{x})$ is provided in literature by the Coons surface [11], $f(x,y)$. This surface satisfies the Dirichlet boundary constraints,

$$f(0,y) = c(0,y), \quad f(1,y) = c(1,y), \quad f(x,0) = c(x,0), \quad \text{and} \quad f(x,1) = c(x,1), \tag{7}$$

where the surface is contained in the $x,y \in [0,1] \times [0,1]$ domain. This surface is used in computer graphics and in computational mechanics applications to smoothly join other surfaces together, particularly in finite element method and boundary element method, to mesh problem domains into elements. The expression of the Coons surface is,

$$\begin{aligned} f(x,y) = {} & (1-x)c(0,y) + x\,c(1,y) + (1-y)\,c(x,0) + y\,c(x,1) - x\,y\,c(1,1) \\ & - (1-x)(1-y)\,c(0,0) - (1-x)\,y\,c(0,1) - x\,(1-y)\,c(1,0), \end{aligned}$$

where the four subtracting terms are there for continuity. Note the constraint functions at boundary corners must have the same value, $c(0,0)$, $c(0,1)$, $c(1,0)$, and $c(1,1)$. This equation can be written in matrix form as,

$$f(x,y) = \left\{ 1, \quad 1-x, \quad x \right\} \begin{bmatrix} 0 & c(x,0) & c(x,1) \\ c(0,y) & -c(0,0) & -c(0,1) \\ c(1,y) & -c(1,0) & -c(1,1) \end{bmatrix} \left\{ \begin{array}{c} 1 \\ 1-y \\ y \end{array} \right\},$$

or, equivalently,

$$f(x,y) = \boldsymbol{v}^{\mathsf{T}}(x)\,\mathcal{M}(c(x,y))\,\boldsymbol{v}(y), \tag{8}$$

where

$$\mathcal{M}(c(x,y)) = \begin{bmatrix} 0 & c(x,0) & c(x,1) \\ c(0,y) & -c(0,0) & -c(0,1) \\ c(1,y) & -c(1,0) & -c(1,1) \end{bmatrix} \quad \text{and} \quad \boldsymbol{v}(z) = \left\{ \begin{array}{c} 1 \\ 1-z \\ z \end{array} \right\}.$$

Since the $f(x,y)$ boundaries match the boundaries of the $c(x,y)$ constraint function, then the identity, $f(x,y) = \boldsymbol{v}^{\mathsf{T}}(x)\,\mathcal{M}(f(x,y))\,\boldsymbol{v}(y)$, holds for *any* $f(x,y)$ function. Therefore, the $B(\boldsymbol{x})$ function can be set as,

$$B(\boldsymbol{x}) := g(x,y) - \boldsymbol{v}^{\mathsf{T}}(x)\,\mathcal{M}(g(x,y))\,\boldsymbol{v}(y), \tag{9}$$

representing all functions that are always zero at the boundary constraints, as $g(x,y)$ is a free function.

## 4. Theory of Connections Surface Subject to Dirichlet Constraints

Equations (8) and (9) can be merged to provide *all surfaces* with the boundary constraints defined in Equation (7) in the following compact form,

$$f(x,y) = \underbrace{\boldsymbol{v}^{\mathsf{T}}(x)\mathcal{M}(c(x,y))\boldsymbol{v}(y)}_{A(x,y)} + \underbrace{g(x,y) - \boldsymbol{v}^{\mathsf{T}}(x)\mathcal{M}(g(x,y))\boldsymbol{v}(y)}_{B(x,y)}. \tag{10}$$

where, again, $A(x,y)$ indicates an expression satisfying the boundary function constraints defined by $c(x,y)$ and $B(x,y)$ an expression that is zero at the boundaries. In matrix form, Equation (10) becomes,

$$f(x,y) = \left\{ \begin{array}{c} 1 \\ 1-x \\ x \end{array} \right\}^{\mathsf{T}} \begin{bmatrix} g(x,y) & c(x,0) - g(x,0) & c(x,1) - g(x,1) \\ c(0,y) - g(0,y) & g(0,0) - c(0,0) & g(0,1) - c(0,1) \\ c(1,y) - g(1,y) & g(1,0) - c(1,0) & g(1,1) - c(1,1) \end{bmatrix} \left\{ \begin{array}{c} 1 \\ 1-y \\ y \end{array} \right\},$$

where $g(x,y)$ is a freely chosen function. In particular, if $g(x,y) = 0$, then the ToC surface becomes the Coons surface.

Figure 5 (left) shows the Coons surface subject to the constraints,

$$c(x,0) = \sin(3x - \pi/4)\, \cos(\pi/3)$$
$$c(x,1) = \sin(3x - \pi/4)\, \cos(4 + \pi/3)$$
$$c(0,y) = \sin(-\pi/4)\, \cos(4y + \pi/3)$$
$$c(1,y) = \sin(3 - \pi/4)\, \cos(4y + \pi/3),$$

and Figure 5 (right) shows a ToC surface that is obtained using the free function,

$$g(x,y) = \frac{1}{3}\cos(4\pi x)\, \sin(6\pi y) - x^2\, \cos(2\pi y). \tag{11}$$

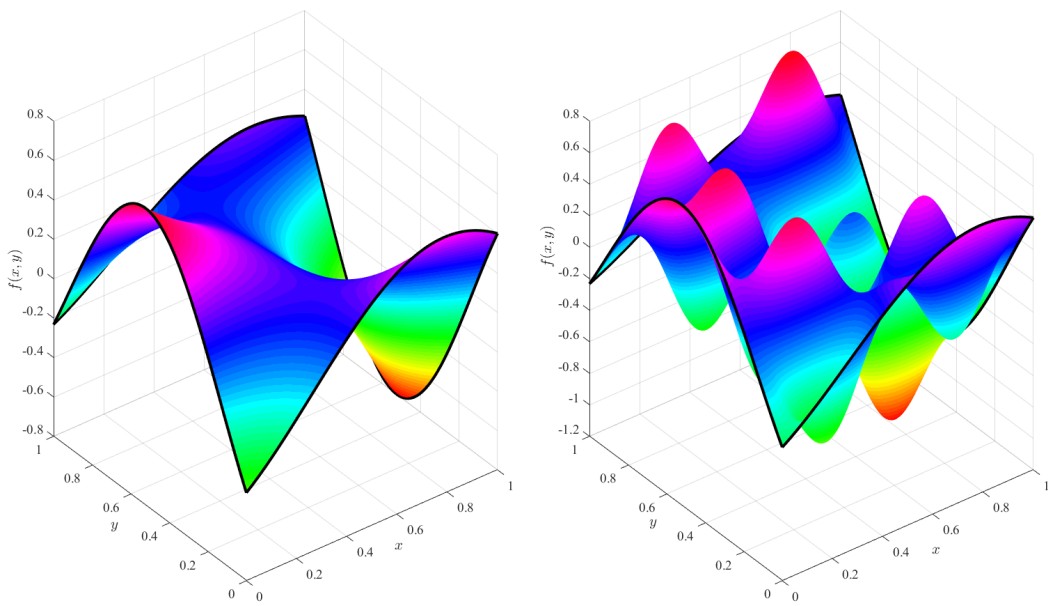

**Figure 5.** Coons surface (**left**); and ToC surface (**right**) using $g(x,y)$ provided in Equation (11).

For generic boundaries defined in the rectangle $x, y \in [x_i, x_f] \times [y_i, y_f]$, the ToC surface becomes,

$$
\begin{aligned}
f(x,y) = g(x,y) &+ \frac{x - x_f}{x_i - x_f}\left[c(x_i,y) - g(x_i,y)\right] + \frac{x - x_i}{x_f - x_i}\left[c(x_f,y) - g(x_f,y)\right] \\
&+ \frac{y - y_f}{y_i - y_f}\left[c(x,y_i) - g(x,y_i)\right] + \frac{y - y_i}{y_f - y_i}\left[c(x,y_f) - g(x,y_f)\right] \\
&- \frac{(x - x_f)(y - y_f)}{(x_i - x_f)(y_i - y_f)}\left[c(x_i,y_i) - g(x_i,y_i)\right] \\
&- \frac{(x - x_f)(y - y_i)}{(x_i - x_f)(y_f - y_i)}\left[c(x_i,y_f) - g(x_i,y_f)\right] \\
&- \frac{(x - x_i)(y - y_f)}{(x_f - x_i)(y_i - y_f)}\left[c(x_f,y_i) - g(x_f,y_i)\right] \\
&- \frac{(x - x_i)(y - y_i)}{(x_f - x_i)(y_f - y_i)}\left[c(x_f,y_f) - g(x_f,y_f)\right].
\end{aligned}
\tag{12}
$$

Equation (12) can also be set in matrix form,

$$f(x,y) = v_x^{\mathrm{T}}(x, x_i, x_f)\, \mathcal{M}(x,y)\, v_y(y, y_i, y_f)$$

where

$$\mathcal{M}(x,y) = \begin{bmatrix} g(x,y) & c(x,y_i)-g(x,y_i) & c(x,y_f)-g(x,y_f) \\ c(x_i,y)-g(x_i,y) & g(x_i,y_i)-c(x_i,y_i) & g(x_i,y_f)-c(x_i,y_f) \\ c(x_f,y)-g(x_f,y) & g(x_f,y_i)-c(x_f,y_i) & g(x_f,y_f)-c(x_f,y_f) \end{bmatrix}$$

and

$$\boldsymbol{v}_x(x,x_i,x_f) = \left\{ \begin{array}{c} 1 \\ \dfrac{x-x_f}{x_i-x_f} \\ \dfrac{x-x_i}{x_f-x_i} \end{array} \right\} \quad \text{and} \quad \boldsymbol{v}_y(y,y_i,y_f) = \left\{ \begin{array}{c} 1 \\ \dfrac{y-y_f}{y_i-y_f} \\ \dfrac{y-y_i}{y_f-y_i} \end{array} \right\}.$$

Note that all the ToC surfaces provided are linear in $g(x,y)$, and, therefore, they can be used to solve, by linear/nonlinear least-squares, two-dimensional optimization problems subject to boundary function constraints, such as linear/nonlinear partial differential equations.

## 5. Multi-Function Constraints at Generic Coordinates

Equation (12) can be generalized to many function constraints (grid of functions). Assume a set of $n_x$ function constraints $c(x_k,y)$ and a set of $n_y$ function constraints $c(x,y_k)$ intersecting at the $n_x n_y$ points $p_{ij} = c(x_i,y_j)$, then all surfaces satisfying the $n_x n_y$ function constraints can be expressed by,

$$\begin{aligned} f(x,y) = g(x,y) &+ \sum_{k=1}^{n_x} [c(x_k,y)-g(x_k,y)] \prod_{i \neq k} \frac{x-x_i}{x_k-x_i} \\ &+ \sum_{k=1}^{n_y} [c(x,y_k)-g(x,y_k)] \prod_{i \neq k} \frac{y-y_i}{y_k-y_i} \\ &- \sum_{i=1}^{n_x} \left\{ \sum_{j=1}^{n_y} \frac{(x-x_j)(y-y_i)}{(x_i-x_j)(y_j-y_i)} [c(x_i,y_j)-g(x_i,y_j)] \right\}. \end{aligned} \tag{13}$$

Again, Equation (13) can be written in compact form,

$$f(x,y) = \boldsymbol{v}^{\mathsf{T}}(x)\,\mathcal{M}(c(x,y))\,\boldsymbol{v}(y) + g(x,y) - \boldsymbol{v}^{\mathsf{T}}(x)\,\mathcal{M}(g(x,y))\,\boldsymbol{v}(y)$$

where,

$$\boldsymbol{v}(x) = \left\{ \begin{array}{c} 1 \\ \prod_{i \neq 1} \dfrac{x-x_i}{x_1-x_i} \\ \vdots \\ \prod_{i \neq n_x} \dfrac{x-x_i}{x_{n_x}-x_i} \end{array} \right\} \quad \text{and} \quad \boldsymbol{v}(y) = \left\{ \begin{array}{c} 1 \\ \prod_{i \neq 1} \dfrac{y-y_i}{y_1-y_i} \\ \vdots \\ \prod_{i \neq n_y} \dfrac{y-y_i}{y_{n_y}-y_i} \end{array} \right\}$$

and

$$\mathcal{M}(c(x,y)) = \begin{bmatrix} 0 & c(x,y_1) & \cdots & c(x,y_{n_y}) \\ c(x_1,y) & -c(x_1,y_1) & \cdots & -c(x_1,y_{N_y}) \\ \vdots & \vdots & \ddots & \vdots \\ c(x_{n_x},y) & -c(x_{n_x},y_1) & \cdots & -c(x_{n_x},y_{n_y}) \end{bmatrix}$$

For example, two function constraints in $x$ and three function constraints in $y$ can be obtained using the matrix,

$$\mathcal{M}(c(x,y)) = \begin{bmatrix} 0 & c(x,y_1) & c(x,y_2) & c(x,y_3) \\ c(x_1,y) & -c(x_1,y_1) & -c(x_1,y_2) & -c(x_1,y_3) \\ c(x_2,y) & -c(x_2,y_1) & -c(x_2,y_2) & -c(x_2,y_3) \end{bmatrix}$$

and the vectors,

$$v(x) = \left\{\begin{array}{c} 1 \\ \dfrac{x - x_2}{x_1 - x_2} \\ \dfrac{x - x_1}{x_2 - x_1} \end{array}\right\} \qquad \text{and} \qquad v(y) = \left\{\begin{array}{c} 1 \\ \dfrac{(y - y_2)(y - y_3)}{(y_1 - y_2)(y_1 - y_3)} \\ \dfrac{(y - y_1)(y - y_3)}{(y_2 - y_1)(y_2 - y_3)} \\ \dfrac{(y - y_2)(y - y_1)}{(y_3 - y_2)(y_3 - y_1)} \end{array}\right\}.$$

Two examples of ToC surfaces are given in Figure 6 in the $x, y \in [-2, 1] \times [1, 3]$ domain.

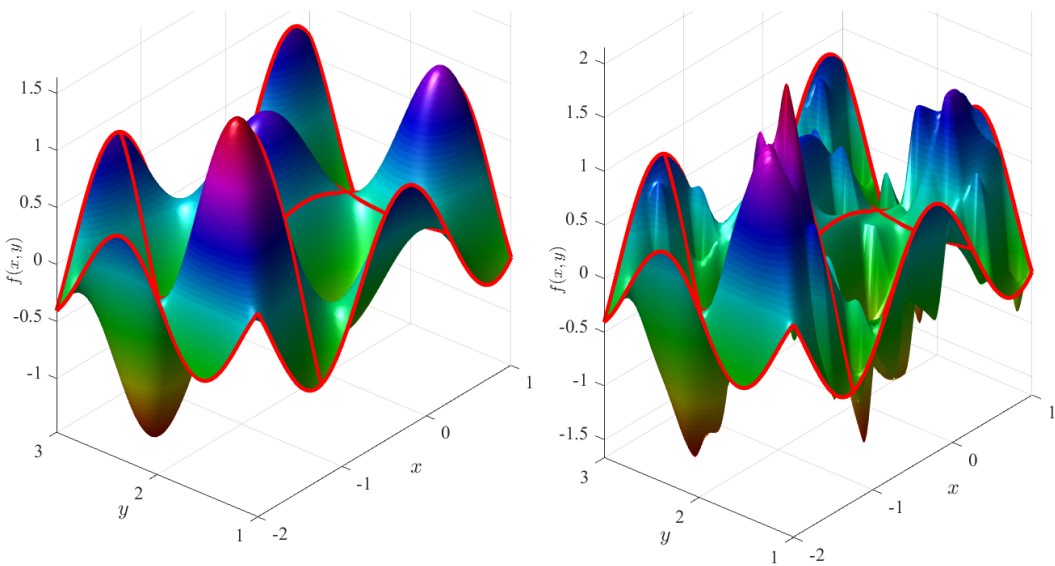

**Figure 6.** ToC surface subject to multiple constraints on two axes: using $g(x,y) = 0$ (**left**); and using $g(x,y) = \mathrm{mod}(x, 0.5)\cos(19y) - x\,\mathrm{mod}(3y, 0.4)$ (**right**).

## 6. Constraints on Function and Derivatives

The "Boolean sum formulation" was provided by Farin [12] (also called "Hermite–Coons formulation") of the Coons surface that includes boundary derivatives,

$$f(x,y) = v^{\mathsf{T}}(y)F^x(x) + v^{\mathsf{T}}(x)F^y(y) - v^{\mathsf{T}}(x)M^{xy}v(y) \tag{14}$$

where

$$v(z) := \{2z^3 - 3z^2 + 1, \quad z^3 - 2z^2 + z, \quad -2z^3 + 3z^2, \quad z^3 - z^2\}^{\mathsf{T}}$$
$$F^x(x) := \{c(x,0), \quad c_y(x,0), \quad c(x,1), \quad c_y(x,1)\}^{\mathsf{T}}$$
$$F^y(y) := \{c(0,y), \quad c_x(0,y), \quad c(1,y), \quad c_x(1,y)\}^{\mathsf{T}}$$

and

$$M^{xy}(x,y) := \begin{bmatrix} c(0,0) & c_y(0,0) & c(0,1) & c_y(0,1) \\ c_x(0,0) & c_{xy}(0,0) & c_x(0,1) & c_{xy}(0,1) \\ c(1,0) & c_y(1,0) & c(1,1) & c_y(1,1) \\ c_x(1,0) & c_{xy}(1,0) & c_x(1,1) & c_{xy}(1,1) \end{bmatrix}.$$

The formulation provided in Equation (14) can be put in the matrix compact form,

$$f(x,y) = v^{\mathsf{T}}(x)\,\mathcal{M}(c(x,y))\,v(y), \tag{15}$$

where

$$v(z) := \{1, \quad 2z^3 - 3z^2 + 1, \quad z^3 - 2z^2 + z, \quad -2z^3 + 3z^2, \quad z^3 - z^2\}^\mathsf{T} \tag{16}$$

and the $5 \times 5$ matrix, $\mathcal{M}(c(x,y))$, has the expression,

$$\mathcal{M}(c(x,y)) := \begin{bmatrix} 0 & c(x,0) & c_y(x,0) & c(x,1) & c_y(x,1) \\ c(0,y) & -c(0,0) & -c_y(0,0) & -c(0,1) & -c_y(0,1) \\ c_x(0,y) & -c_x(0,0) & -c_{xy}(0,0) & -c_x(0,1) & -c_{xy}(0,1) \\ c(1,y) & -c(1,0) & -c_y(1,0) & -c(1,1) & -c_y(1,1) \\ c_x(1,y) & -c_x(1,0) & -c_{xy}(1,0) & -c_x(1,1) & -c_{xy}(1,1) \end{bmatrix}. \tag{17}$$

To verify the boundary derivative constraints, the following partial derivatives of Equation (15) are used,

$$f_x(x,y) = [v_x^\mathsf{T}(x)\mathcal{M}(c(x,y)) + v^\mathsf{T}(x)\mathcal{M}_x(c(x,y))]v(y)$$
$$f_y(x,y) = v^\mathsf{T}(x)[\mathcal{M}_y^\mathsf{T}(c(x,y))v(y) + \mathcal{M}(c(x,y))v_y(y)],$$

where

$$\frac{dv}{dz} = \begin{Bmatrix} 0 \\ 6z(z-1) \\ 3z^2 - 4z + 1 \\ 6z(1-z) \\ z(3z-2) \end{Bmatrix}, \quad \mathcal{M}_y = \begin{bmatrix} 0 & \mathbf{0}_{1\times4} \\ c_y(0,y) & \mathbf{0}_{1\times4} \\ c_{xy}(0,y) & \mathbf{0}_{1\times4} \\ c_y(1,y) & \mathbf{0}_{1\times4} \\ c_{xy}(1,y) & \mathbf{0}_{1\times4} \end{bmatrix}, \quad \text{and} \quad \mathcal{M}_x^\mathsf{T} = \begin{bmatrix} 0 & \mathbf{0}_{1\times4} \\ c_x(x,0) & \mathbf{0}_{1\times4} \\ c_{xy}(x,0) & \mathbf{0}_{1\times4} \\ c_x(x,1) & \mathbf{0}_{1\times4} \\ c_{xy}(x,1) & \mathbf{0}_{1\times4} \end{bmatrix}.$$

The ToC in 2D with function and derivative boundary constraints is simply,

$$f(x,y) = \underbrace{v^\mathsf{T}(x)\mathcal{M}(c(x,y))v(y)}_{A(x,y)} + \underbrace{g(x,y) - v^\mathsf{T}(x)\mathcal{M}(g(x,y))v(y)}_{B(x,y)} \tag{18}$$

where the $\mathcal{M}$ matrix and the $v$ vectors are provided by Equations (17) and (16), respectively.

Dirichlet/Neumann mixed constraints can be derived, as shown in the examples provided in Sections 6.1–6.4. The matrix compact form is simply obtained from the matrix defined in Equation (17) by removing the rows and the columns associated with the boundary constraints not provided, while the vectors $v(x)$ and $v(y)$ are derived by specifying the constraints. Note that in general the vectors $v(x)$ and $v(y)$ are *not* unique. The reason the vectors $v(x)$ and $v(y)$ are not unique comes from the fact that the $A(x)$ term in Equation (6) is not unique.

In the next subsections, four Dirichlet/Neumann mixed constraint examples providing the simplest expressions for $v(x)$ and $v(y)$ are derived. The Appendix A contains the expressions for the $v(x)$ and $v(y)$ vectors associated with all the combinations of Dirichlet and Neumann constraints.

### 6.1. Constraints: $c(0,y)$ and $c(x,0)$

In this case, the Coons-type surface satisfying the boundary constraints can be expressed as,

$$f(x,y) = \begin{Bmatrix} 1 & p(x) \end{Bmatrix} \begin{bmatrix} 0 & c(x,0) \\ c(0,y) & -c(0,0) \end{bmatrix} \begin{Bmatrix} 1 \\ q(y) \end{Bmatrix}$$

where $p(x)$ and $q(y)$ are unknown functions. Expanding, we obtain $f(x,y) = c(x,0)q(y) + p(x)[c(0,y) - c(0,0)q(y)]$. The two constraints are satisfied if,

$$c(0,y) = c(0,0)q(y) + p(0)[c(0,y) - c(0,0)q(y)]$$
$$c(x,0) = c(x,0)q(0) + p(x)[c(0,0) - c(0,0)q(0)].$$

Therefore, the $p(x)$ and $q(y)$ functions must satisfy $p(0) = 1$ and $q(0) = 1$. The simplest expressions satisfying these equations can be obtained by selecting $p(x) = 1$ and $q(y) = 1$. In this case, the associated ToC surface is given by,

$$f(x,y) = \begin{Bmatrix} 1 & 1 \end{Bmatrix} \begin{bmatrix} g(x,y) & c(x,0) - g(x,0) \\ c(0,y) - g(0,y) & g(0,0) - c(0,0) \end{bmatrix} \begin{Bmatrix} 1 \\ 1 \end{Bmatrix}$$

Note that any functions satisfying $p(0) = 1$ and $q(0) = 1$ can be adopted to obtain the ToC surface satisfying the constraints $f(0,y) = c(0,y)$ and $f(x,0) = c(x,0)$. This is because there are infinite Coons-type surfaces satisfying the constraints. Consequently, the vectors $v(x)$ and $v(y)$ are not unique.

### 6.2. Constraints: $c(0,y)$ and $c_y(x,0)$

For these boundary constraints, the Coons-type surface is expressed by,

$$f(x,y) = \begin{Bmatrix} 1 & p(x) \end{Bmatrix} \begin{bmatrix} 0 & c_y(x,0) \\ c(0,y) & -c_y(0,0) \end{bmatrix} \begin{Bmatrix} 1 \\ q(y) \end{Bmatrix}$$
$$= c_y(x,0)q(y) + p(x)[c(0,y) - c_y(0,0)q(y)].$$

The constraints are satisfied if,

$$c(0,y) = c_y(0,0)q(y) + p(0)[c(0,y) - c_y(0,0)q(y)],$$
$$c_y(x,0) = c_y(x,0)q_y(0) + p(x)[c_y(0,0) - c_y(0,0)q_y(0)].$$

Therefore, the $p(x)$ and $q(y)$ functions must satisfy $p(0) = 1$ and $q_y(0) = 1$. One solution is $p(x) = 1$ and $q(y) = y$. Therefore, the associated ToC surface is given by,

$$f(x,y) = \begin{Bmatrix} 1 & 1 \end{Bmatrix} \begin{bmatrix} g(x,y) & c_y(x,0) - g_y(x,0) \\ c(0,y) - g(0,y) & g_y(0,0) - c_y(0,0) \end{bmatrix} \begin{Bmatrix} 1 \\ y \end{Bmatrix}.$$

### 6.3. Neumann Constraints: $c_x(0,y)$, $c_x(1,y)$, $c_y(x,0)$, and $c_y(x,1)$

In this case, the Coons-type surface satisfying the boundary constraints can be expressed as,

$$f(x,y) = \begin{Bmatrix} 1, & p_1(x), & p_2(x) \end{Bmatrix} \begin{bmatrix} 0 & c_y(x,0) & c_y(x,1) \\ c_x(0,y) & -c_{xy}(0,0) & -c_{xy}(0,1) \\ c_x(1,y) & -c_{xy}(1,0) & -c_{xy}(1,1) \end{bmatrix} \begin{Bmatrix} 1 \\ q_1(y) \\ q_2(y) \end{Bmatrix}.$$

The constraints are satisfied if,

$$
\begin{aligned}
c_x(0,y) = {} & q_1(y)c_{xy}(0,0) + q_2(y)c_{xy}(0,1) + \\
& + p_{1x}(0)[c_x(0,y) - q_1(y)c_{xy}(0,0) - q_2(y)c_{xy}(0,1)] + \\
& + p_{2x}(0)[c_x(1,y) - q_1(y)c_{xy}(1,0) - q_2(y)c_{xy}(1,1)] \\
c_x(1,y) = {} & q_1(y)c_{xy}(1,0) + q_2(y)c_{xy}(1,1) + \\
& + p_{1x}(1)[c_x(0,y) - q_1(y)c_{xy}(0,0) - q_2(y)c_{xy}(0,1)] + \\
& + p_{2x}(1)[c_x(1,y) - q_1(y)c_{xy}(1,0) - q_2(y)c_{xy}(1,1)] \\
c_y(x,0) = {} & q_{1y}(0)c_y(x,0) + q_{2y}(0)c_y(x,1) + \\
& + p_1(x)[c_{xy}(0,0) - q_{1y}(0)c_{xy}(0,0) - q_{2y}(0)c_{xy}(0,1)] + \\
& + p_2(x)[c_{xy}(1,0) - q_{1y}(0)c_{xy}(1,0) - q_{2y}(0)c_{xy}(1,1)] \\
c_y(x,1) = {} & q_{1y}(1)c_y(x,0) + q_{2y}(1)c_y(x,1) + \\
& + p_1(x)[c_{xy}(0,1) - q_{1y}(1)c_{xy}(0,0) - q_{2y}(1)c_{xy}(0,1)] + \\
& + p_2(x)[c_{xy}(1,1) - q_{1y}(1)c_{xy}(1,0) - q_{2y}(1)c_{xy}(1,1)].
\end{aligned}
$$

These equations imply $p_{1x}(0) = q_{1x}(0) = 1$, $p_{1x}(1) = q_{1x}(1) = 0$, $p_{2x}(0) = q_{2x}(0) = 0$, and $p_{2x}(1) = q_{2x}(1) = 1$. Therefore, the simplest solution is $p_1(t) = q_1(t) = t - t^2/2$ and $p_2(t) = q_2(t) = t^2/2$. Then, the associated ToC surface satisfying the Neumann constraints is given by,

$$
f(x,y) = v^{\mathsf{T}}(x)
\begin{bmatrix}
g(x,y) & c_y(x,0) - g_y(x,0) & c_y(x,1) - g_y(x,1) \\
c_x(0,y) - g_x(0,y) & g_{xy}(0,0) - c_{xy}(0,0) & g_{xy}(0,1) - c_{xy}(0,1) \\
c_x(1,y) - g_x(1,y) & g_{xy}(1,0) - c_{xy}(1,0) & g_{xy}(1,1) - c_{xy}(1,1)
\end{bmatrix}
v(y)
$$

where

$$
v^{\mathsf{T}}(x) = \left\{ 1, \quad x - \frac{x^2}{2}, \quad \frac{x^2}{2} \right\}
\qquad \text{and} \qquad
v(y) = \left\{ 1, \quad y - \frac{y^2}{2}, \quad \frac{y^2}{2} \right\}.
$$

*6.4. Constraints: $c(0,y)$, $c_y(x,0)$, and $c_y(x,1)$*

In this case, the Coons-type surface satisfying the boundary constraints is in the form,

$$
f(x,y) = \left\{ \begin{matrix} 1 \\ p(x) \end{matrix} \right\}^{\mathsf{T}}
\begin{bmatrix}
0 & c_y(x,0) & c_y(x,1) \\
c(0,y) & -c_y(0,0) & -c_y(0,1)
\end{bmatrix}
\left\{ \begin{matrix} 1 \\ q_1(y) \\ q_2(y) \end{matrix} \right\}.
$$

The constraints are satisfied if $p(0) = 1$, $p_{1y}(0) = 1$, $p_{1y}(1) = 0$, $p_{2y}(0) = 0$, and $p_{2y}(1) = 1$. Therefore, the associated ToC surface is,

$$
f(x,y) = \left\{ \begin{matrix} 1 \\ 1 \end{matrix} \right\}^{\mathsf{T}}
\begin{bmatrix}
g(x,y) & c_y(x,0) - g_y(x,0) & c_y(x,1) - g_y(x,1) \\
c(0,y) - g(0,y) & g_y(0,0) - c_y(0,0) & g_y(0,1) - c_y(0,1)
\end{bmatrix}
\left\{ \begin{matrix} 1 \\ y - \dfrac{y^2}{2} \\ \dfrac{y^2}{2} \end{matrix} \right\}.
$$

*6.5. Generic Mixed Constraints*

Consider the case of mixed constraints,

$$
\begin{array}{lll}
f(x,y_1) = c(x,y_1) & & f_y(x_1,y) = c_y(x_1,y) \\
f_x(x,y_2) = c_x(x,y_2) & \text{and} & f_y(x_2,y) = c_y(x_2,y) \, . \\
f(x,y_3) = c(x,y_3) & & f(x_3,y) = c(x_3,y)
\end{array}
\tag{19}
$$

In this case, the surface satisfying the boundary constraints is built using the matrix,

$$
\mathcal{M}(c(x,y)) = \begin{bmatrix} 0 & c(x,y_1) & c_x(x,y_2) & c(x,y_3) \\ c_y(x_1,y) & -c_y(x_1,y_1) & -c_{xy}(x_1,y_2) & -c_y(x_1,y_3) \\ c_y(x_2,y) & -c_y(x_2,y_1) & -c_{xy}(x_2,y_2) & -c_y(x_2,y_3) \\ c(x_3,y) & -c(x_3,y_1) & -c_x(x_3,y_2) & -c(x_3,y_3) \end{bmatrix}
$$

and all surfaces subject to the constraints defined in Equation (19) can be obtained by,

$$
f(x,y) = v(x)^{\mathsf{T}}\mathcal{M}(c(x,y))v(y) + g(x,y) - v(x)^{\mathsf{T}}\mathcal{M}(g(x,y))v(y),
$$

where

$$
v(x) = \begin{Bmatrix} 1 \\ p_1(x,x_1,x_2,x_3) \\ p_2(x,x_1,x_2,x_3) \\ p_3(x,x_1,x_2,x_3) \end{Bmatrix} \quad \text{and} \quad v(y) = \begin{Bmatrix} 1 \\ q_1(y,y_1,y_2,y_3) \\ q_2(y,y_1,y_2,y_3) \\ q_3(y,y_1,y_2,y_3) \end{Bmatrix}
$$

are vectors made of the (not unique) function vectors $v(x)$ and $v(y)$ whose expressions can be found by satisfying the constraints (as done in the previous four subsections) along with a methodology similar to that given in Section 5.

## 7. Extension to *n*-Dimensional Spaces and Arbitrary-Order Derivative Constraints

This section provides the *Multivariate Theory of Connections*, as the generalization to *n*-dimensional rectangular domains with arbitrary-order boundary derivatives of what is presented above for two-dimensional space. Using tensor notation, this generalization is represented in the following compact form,

$$
F(x) = \underbrace{\mathcal{M}(c(x))_{i_1 i_2 \ldots i_n}\, v_{i_1}\, v_{i_2} \ldots v_{i_n}}_{A(x)} + \underbrace{g(x) - \mathcal{M}(g(x))_{i_1 i_2 \ldots i_n}\, v_{i_1}\, v_{i_2} \ldots v_{i_n}}_{B(x)} \tag{20}
$$

where $n$ is the number of orthogonal coordinates defined by the vector $x = \{x_1, x_2, \ldots, x_n\}$, $v_{i_k}(x_k)$ is the $i_k$th element of a vector function of the variable $x_k$, $\mathcal{M}$ is an $n$-dimensional tensor that is a function of the boundary constraints defined in $c(x)$, and $g(x)$ is the free-function.

In Equation (20), the term $A(x)$ represents any function satisfying the boundary constraints defined by $c(x)$ and the term $B(x)$ represents all possible functions that are zero on the boundary constraints. The subsections that follow explain how to construct the $\mathcal{M}$ tensor and the $v_{i_k}$ vectors for assigned boundary constraints, and provides a proof that the tensor formulation of the ToC defined by Equation (20) satisfies all boundary constraints defined by $c(x)$, independently of the choice of the free function, $g(x)$.

Consider a generic boundary constraint on the $x_k = p$ hyperplane, where $k \in [1, n]$. This constraint specifies the $d$-derivative of the constraint function $c(x)$ evaluated at $x_k = p$ and it is indicated by $^{k}c_p^d := \dfrac{\partial^d c(x)}{\partial x_k^d}\bigg|_{x_k=p}$. Consider a set of $\ell_k$ constraints defined in various $x_k$ hyperplanes. This set of constraints is indicated by $^{k}c_{p^k}^{d^k}$, where $d^k$ and $p^k$ are vectors of $\ell_k$ elements indicating the order of derivatives and the values of $x_k$ where the boundary constraints are defined, respectively. A specific boundary constraint, e.g. the $m$th boundary constraint, can then be written as $^{k}c_{p_m^k}^{d_m^k}$.

Additionally, let us define an operator, called the boundary constraint operator, whose purpose is to take the $d$th derivative with respect to coordinate $x_k$ and then evaluate that function at $x_k = p$. Equation (21) shows the idea.

$$
^{k}b_p^d[f] \equiv \frac{\partial^d f}{\partial x_k^d}\bigg|_{(x_1,\ldots,x_{k-1},p,x_{k+1},\ldots,x_n)} \tag{21}
$$

In general, for a function of $n$ variables, the boundary constraint operator identifies an $n-1$-dimensional manifold. As the boundary constraint operator is used throughout this proof, it is important to note its properties when acting on sums and products of functions. Equation (22) shows how the boundary constraint operator acts on sums, and Equation (23) shows how the boundary constraint operator acts on products.

$$^k b_p^d[f_1 + f_2] = {}^k b_p^d[f_1] + {}^k b_p^d[f_2] \tag{22}$$

$$^k b_p^d[f_1 f_2] = \begin{cases} {}^k b_p^d[f_1] \, {}^k b_p^d[f_2], & d = 0 \\ {}^k b_p^d[f_1] f_2 + f_1 \, {}^k b_p^d[f_2], & d > 0 \end{cases} \tag{23}$$

This section shows how to build the $\mathcal{M}$ tensor and the vectors $v$ given the boundary constraints defined by the boundary constraint operators. Moreover, this section contains a proof that, in Equation (20), the boundary constraints defined by $c(x)$ satisfy the function $A(x)$ and, by extension, the function $B(x)$ projects the free-function $g(x)$ onto the sub-space of functions that are zero on the boundary constraints. Then, it follows that the expression for the ToC surface given in Equation (20) represents *all* possible functions that meet the boundary defined by the boundary constraint operators.

### 7.1. The $\mathcal{M}$ Tensor

There is a step-by-step method for constructing the $\mathcal{M}$ tensor.

1. The element of $\mathcal{M}$ for all indices equal to 1 is 0 (i.e., $\mathcal{M}_{11\ldots1} = 0$).
2. The first order tensor obtained by keeping the $k$th dimension's index and setting all other dimension's indices to 1 can be written as,

$$\mathcal{M}_{1,\ldots,1,i_k,1,\ldots,1} = {}^k c_{p^k}^{d^k}, \qquad \text{where} \quad i_k \in [2, \ell_k + 1],$$

where the vector $^k c_{p^k}^{d^k}$ contains the $\ell_k$ boundary constraints specified along the $x_k$-axis. For example, consider the following $\ell_7 = 3$ constraints on the $k = 7$th axis,

$$^7 c_{p^7}^{d^7} := \left\{ c|_{x_7 = -0.3}, \quad \frac{\partial^4 c}{\partial x_7^4}\bigg|_{x_7 = 0.5}, \quad \frac{\partial c}{\partial x_7}\bigg|_{x_7 = 1.1} \right\} \qquad \text{then}: \begin{cases} d^7 = \{0,\, 4,\, 1\} \\ p^7 = \{-0.3,\, 0.5,\, 1.1\}. \end{cases}$$

3. The generic element of the tensor is $\mathcal{M}_{i_1 i_2 \ldots i_n}$, where at least two indices are different from 1. Let $m$ be the number of indices different from 1. Note that $m$ is also the number of constraint "intersections". In this case, the generic element of the $\mathcal{M}$ tensor is provided by,

$$\mathcal{M}_{i_1 i_2 \ldots i_n} = {}^1 b_{p_{i_1-1}^1}^{d_{i_1-1}^1} \left[ {}^2 b_{p_{i_2-1}^2}^{d_{i_2-1}^2} \left[ \ldots \left[ {}^n b_{p_{i_n-1}^n}^{d_{i_n-1}^n} [c(x)] \right] \ldots \right] \right] (-1)^{m+1}. \tag{24}$$

If $c(x) \in C^s$, where $s = \sum_{k=1}^{n} d_{i_k-1}^k$, then Clairaut's theorem states that the sequence of boundary constraint operators provided in Equation (24) can be freely permutated. This permutation becomes obvious by multiple applications of the theorem. For example,

$$f_{xyy} = (f_{xy})_y = (f_{yx})_y = (f_y)_{xy} = (f_y)_{yx} = f_{yyx}.$$

To better clarify how to use Equation (24), consider the example of the following constraints in three-dimensional space.

$$c(x)|_{x_1=0}, \quad c(x)|_{x_1=1}, \quad c(x)|_{x_2=0}, \quad \frac{\partial c(x)}{\partial x_2}\bigg|_{x_2=0}, \quad c(x)|_{x_3=0}, \quad \text{and} \quad \frac{\partial c(x)}{\partial x_3}\bigg|_{x_3=0}$$

1. From Step 1: $M_{111} = 0$
2. From Step 2:

$$M_{i_1 11} = \left\{ 0, \quad c(0, x_2, x_3), \quad c(1, x_2, x_3) \right\} = \left\{ 0, \quad {}^1b_0^0[c(\boldsymbol{x})], \quad {}^1b_1^0[c(\boldsymbol{x})] \right\}$$

$$M_{1i_2 1} = \left\{ 0, \quad c(x_1, 0, x_3), \quad \frac{\partial c}{\partial x_2}(x_1, 0, x_3) \right\} = \left\{ 0, \quad {}^2b_0^0[c(\boldsymbol{x})], \quad {}^2b_0^1[c(\boldsymbol{x})] \right\}$$

$$M_{11i_3} = \left\{ 0, \quad c(x_1, x_3, 0), \quad \frac{\partial c}{\partial x_3}(x_1, x_2, 0) \right\} = \left\{ 0, \quad {}^3b_0^0[c(\boldsymbol{x})], \quad {}^3b_0^1[c(\boldsymbol{x})] \right\}$$

3. From Step 3, a single example is provided,

$$\mathcal{M}_{323} = {}^1b_1^0 \left[ {}^2b_0^0 \left[ {}^3c_0^1(\boldsymbol{x}) \right] \right] (-1)^4 = \left. \frac{\partial c(\boldsymbol{x})}{\partial x_3} \right|_{\substack{x_1=1 \\ x_2=0 \\ x_3=0}}$$

which, thanks to Clairaut's theorem, can also be written as,

$$\mathcal{M}_{323} = {}^2b_0^0 \left[ {}^3b_0^1 \left[ {}^1c_1^0 \right] \right] (-1)^4 = {}^3b_0^1 \left[ {}^1b_1^0 \left[ {}^2c_0^0 \right] \right] (-1)^4.$$

Three additional examples are given to help further illustrate the procedure,

$$M_{132} = - \left. \frac{\partial c(\boldsymbol{x})}{\partial x_2} \right|_{\substack{x_2=0 \\ x_3=0}}, \qquad M_{221} = -c(0, 0, x_3), \qquad \text{and} \qquad M_{333} = \left. \frac{\partial^2 c(\boldsymbol{x})}{\partial x_2 \partial x_3} \right|_{\substack{x_1=1 \\ x_2=0 \\ x_3=0}}$$

*7.2. The* **v** *Vectors*

Each vector, $v_k$, is associated with the $\ell_k$ constraints that are specified by ${}^k c_{\boldsymbol{p}^k}^{\boldsymbol{d}^k}$. The $v_k$ vector is built as follows,

$$v_k = \left\{ 1, \quad \sum_{i=1}^{\ell_k} \alpha_{i1} h_i(x_k), \quad \sum_{i=1}^{\ell_k} \alpha_{i2} h_i(x_k), \quad \ldots, \quad \sum_{i=1}^{\ell_k} \alpha_{i\ell_k} h_i(x_k) \right\},$$

where $h_i(x_k)$ are $\ell_k$ linearly independent functions. The simplest set of linearly independent functions are monomials, that is, $h_i(x_k) = x_k^{i-1}$. The $\ell_k \times \ell_k$ coefficients, $\alpha_{ij}$, can be computed by matrix inversion,

$$\begin{bmatrix} {}^k b_{p_1}^{d_1}[h_1] & {}^k b_{p_1}^{d_1}[h_2] & \cdots & {}^k b_{p_1}^{d_1}[h_{\ell_k}] \\ {}^k b_{p_2}^{d_2}[h_1] & {}^k b_{p_2}^{d_2}[h_2] & \cdots & {}^k b_{p_2}^{d_2}[h_{\ell_k}] \\ \vdots & \vdots & \ddots & \vdots \\ {}^k b_{p_{\ell_k}}^{d_{\ell_k}}[h_1] & {}^k b_{p_{\ell_k}}^{d_{\ell_k}}[h_2] & \cdots & {}^k b_{p_{\ell_k}}^{d_{\ell_k}}[h_{\ell_k}] \end{bmatrix} \begin{bmatrix} \alpha_{11} & \alpha_{12} & \cdots & \alpha_{1\ell_k} \\ \alpha_{21} & \alpha_{22} & \cdots & \alpha_{2\ell_k} \\ \vdots & \vdots & \ddots & \vdots \\ \alpha_{\ell_k 1} & \alpha_{\ell_k 2} & \cdots & \alpha_{\ell_k \ell_k} \end{bmatrix} = \begin{bmatrix} 1 & 0 & \cdots & 0 \\ 0 & 1 & \cdots & 0 \\ \vdots & \vdots & \ddots & \vdots \\ 0 & 0 & \cdots & 1 \end{bmatrix}. \tag{25}$$

To supplement the above explanation, let us look at the example of Dirichlet boundary conditions on $x_1$ from the example in Section 7.1. There are two boundary conditions, $c(\boldsymbol{x})|_{x_1=0}$ and $c(\boldsymbol{x})|_{x_1=1}$, and thus two linearly independent functions are needed,

$$v_{i_1} = \left\{ 1, \quad \alpha_{11} h_1(x_1) + \alpha_{21} h_2(x_1), \quad \alpha_{12} h_1(x_1) + \alpha_{22} h_2(x_1) \right\}.$$

Let us consider, $h_1(x_1) = 1$ and $h_2(x_1) = x_1$. Then, following Equation (25),

$$\begin{bmatrix} {}^1b_0^0[1] & {}^1b_0^0[x] \\ {}^2b_1^0[1] & {}^2b_1^0[x] \end{bmatrix} \begin{bmatrix} \alpha_{11} & \alpha_{12} \\ \alpha_{21} & \alpha_{22} \end{bmatrix} = \begin{bmatrix} 1 & 0 \\ 1 & 1 \end{bmatrix} \begin{bmatrix} \alpha_{11} & \alpha_{12} \\ \alpha_{21} & \alpha_{22} \end{bmatrix} = \begin{bmatrix} 1 & 0 \\ 0 & 1 \end{bmatrix} \quad \rightarrow \quad \begin{bmatrix} \alpha_{11} & \alpha_{12} \\ \alpha_{21} & \alpha_{22} \end{bmatrix} = \begin{bmatrix} 1 & 0 \\ -1 & 1 \end{bmatrix},$$

and substituting the values of $\alpha_{ij}$, we obtain $v_{i_1} = \left\{1, \quad 1 - x_1, \quad x_1\right\}$.

*7.3. Proof*

This section demonstrates that the term $A(x)$ from Equation (20) generates a surface satisfying the boundary constraints defined by the function $c(x)$. First, it is shown that $A(x)$ satisfies boundary constraints on the value, and then that $A(x)$ satisfies boundary constraints on arbitrary-order derivatives.

Equation (23) for $d = 0$ allows us to write,

$$^k b^0_{p_{q-1}}[A(x)] = \, ^k b^0_{p_{q-1}}[\mathcal{M}_{i_1 i_2 \ldots i_k \ldots i_n}]v_{i_1}v_{i_2}\ldots \, ^k b^0_{p_{q-1}}[v_{i_k}]\ldots v_{i_n}. \tag{26}$$

The boundary constraint operator applied to $v_k$ yields,

$$^k b^0_{p_{q-1}}[v_{i_k}] = \begin{cases} = 1, & i_k = 1, q \\ = 0, & i_k \neq 1, q. \end{cases} \tag{27}$$

Since the only nonzero terms are associated with $i_k = 1, q$, we have,

$$^k b^0_{p_{q-1}}[A(x)] = \left( \, ^k b^0_{p_{q-1}}[\mathcal{M}_{i_1 i_2 \ldots 1 \ldots i_n}] + \, ^k b^0_{p_{q-1}}[\mathcal{M}_{i_1 i_2 \ldots q \ldots i_n}] \right) v_{i_1} v_{i_2} \ldots v_{i_n}. \tag{28}$$

Applying the boundary constraint operator to the $n - 1$-dimensional $\mathcal{M}$ tensor where index $i_k = q$ has no effect, because all of the functions already have coordinate $x_k$ substituted for the value $p_{q-1}$ (see Equation (24)). Moreover, applying the boundary constraint operator to the $\mathcal{M}$ tensor where index $i_k = 1$ causes all terms in the sum within the parenthesis in Equation (28) to cancel each other, except when all of the non-$i_k$ indices are equal to one. This leads to Equation (29).

$$^k b^0_{p_{q-1}}[A(x)] = \left( \mathcal{M}_{11\ldots 1 \ldots 1} + \mathcal{M}_{11 \ldots q \ldots 1} \right) v_1 v_1 \ldots v_1 \tag{29}$$

Since $v_j = 1$ when $j = 1$ and $\mathcal{M}_{11\ldots 1} = 0$ by definition, then,

$$^k b^0_{p_{q-1}}[A(x)] = \mathcal{M}_{11\ldots q \ldots 1} = c(x_1, x_2, \ldots, p_{q-1}, \ldots, x_n),$$

which proves Equation (20) works for boundary constraints on the value.

Now, we show that Equation (20) holds for arbitrary-order derivative type boundary constraints. Equation (23) for $d > 0$ allows us to write,

$$^k b^{d_{q-1}}_{p_{q-1}}[A(x)] = \, ^k b^{d_{q-1}}_{p_{q-1}}[\mathcal{M}_{i_1 i_2 \ldots i_k \ldots i_n}]v_{i_1}v_{i_2}\ldots v_{i_k}\ldots v_{i_n} + \mathcal{M}_{i_1 i_2 \ldots i_k \ldots i_n}v_{i_1}v_{i_2}\ldots \, ^k b^{d_{q-1}}_{p_{q-1}}[v_{i_k}]\ldots v_{i_n}. \tag{30}$$

From Equation (23), we note that boundary constraint operators that take a derivative follow the usual product rule when applied to a product. Moreover, we note that all of the $v$ vectors except $v_{i_k}$ do not depend on $x_k$, thus applying the boundary constraint operator to them results in a vector of zeros. Applying the boundary constraint operator to $v_{i_k}$ yields,

$$^k b^{d_{q-1}}_{p_{q-1}}[v_{i_k}] = \begin{cases} = 1, & i_k = q \\ = 0, & i_k \neq q, \end{cases}$$

and applying the boundary constraint operator to $\mathcal{M}$ yields,

$$^{k}b_{p_{q-1}}^{d_{q-1}}[\mathcal{M}_{i_1 i_2 \ldots 1 \ldots i_n}] = \begin{cases} = {}^{k}b_{p_{q-1}}^{d_{q-1}}[\mathcal{M}_{i_1 i_2 \ldots 1 \ldots i_n}], & i_k = 1 \\ = 0, & i_k \neq 1. \end{cases}$$

Substituting these simplifications into $A(x) = \mathcal{M}_{i_1 i_2 \ldots i_k \ldots i_n} v_{i_1} v_{i_2} \ldots v_{i_k} \ldots v_{i_n}$, after applying the boundary constraint operator, results in Equation (31).

$$^{k}b_{p_{q-1}}^{d_{q-1}}[A(x)] = \left( {}^{k}b_{p_{q-1}}^{d_{q-1}}[\mathcal{M}_{i_1 i_2 \ldots 1 \ldots i_n}] + \mathcal{M}_{i_1 i_2 \ldots q \ldots i_n} \right) v_{i_1} v_{i_2} \ldots v_{i_n} \tag{31}$$

Similar to the proof for value-based boundary constraints, based on Equation (24), all terms in the sum within the parenthesis in Equation (31) cancel each other, except when all of the non-$i_k$ indices are equal to one. Thus, Equation (31) can be simplified to Equation (32).

$$^{k}b_{p_{q-1}}^{d_{q-1}}[A(x)] = \left( {}^{k}b_{p_{q-1}}^{d_{q-1}}[\mathcal{M}_{11 \ldots 1 \ldots 1}] + \mathcal{M}_{11 \ldots q \ldots 1} \right) v_1 v_1 \ldots v_1 \tag{32}$$

Again, all of the vectors $v$ were designed such that their first component is 1, and the value of the element of $\mathcal{M}$ for all indices equal to 1 is 0. Therefore, Equation (32) simplifies to,

$$^{k}b_{p_{q-1}}^{d_{q-1}}[A(x)] = \mathcal{M}_{11 \ldots q \ldots 1} = \left. \frac{\partial^d c(x)}{\partial x_k^d} \right|_{x_k = p_{q-1}},$$

which proves Equation (20) works for arbitrary-order derivative boundary constraints.

In conclusion, the term $A(x)$ from Equation (20) generates a manifold satisfying the boundary constraints given in terms of arbitrary-order derivative in $n$-dimensional space. The term $B(x)$ from Equation (20) projects any free function $g(x)$ onto the space of functions that are vanishing at the specified boundary constraints. As a result, Equation (20) can be used to produce the family of *all* possible functions satisfying assigned boundary constraints (functions or derivatives) in rectangular domains in $n$-dimensional space.

## 8. Conclusions

This paper extends to $n$-dimensional spaces the Univariate Theory of Connections (ToC), introduced in Ref. [1]. First, it provides a mathematical tool to express *all* possible surfaces subject to constraint functions and arbitrary-order derivatives in a boundary rectangular domain, and then it extends the results to the multivariate case by providing the Multivariate Theory of Connections, which allows one to obtain $n$-dimensional manifolds subject to any-order derivative boundary constraints.

In particular, if the constraints are provided along one axis only, then this paper shows that the univariate ToC, as defined in Ref. [1], can be adopted to describe *all* possible surfaces satisfying the constraints. If the boundary constraints are defined in a rectangular domain, then the constrained expression is found in the form $f(x) = A(x) + B(x)$, where $A(x)$ can be *any* function satisfying the constraints and $B(x)$ describes *all* functions that are vanishing at the constraints. This is obtained by introducing a free function, $g(x)$, into the function $B(x)$ in such a way that $B(x)$ is zero at the constraints no matter what the $g(x)$ is. This way, by spanning all possible $g(x)$ surfaces (even discontinuous, null, or piece-wise defined) the resulting $B(x)$ generates *all* surfaces that are zero at the constraints and, consequently, $f(x) = A(x) + B(x)$, describes all surfaces satisfying the constraints defined in the rectangular boundary domain. The function $A(x)$ has been selected as a Coons surface [11] and, in particular, a Coons surface is obtained if $g(x) = 0$ is selected. All possible combinations of Dirichlet *and* Neumann constraints are also provided in Appendix A.

The last section provides the Multivariate Theory of Connections extension, which is a mathematical tool to transform $n$-dimensional constraint optimization problems subject to constraints on the boundary value and any-order derivative into unconstrained optimization problems. The number of applications of the Multivariate Theory of Connections are many, especially in the area of partial and stochastic differential equations: the main subjects of our current research.

**Author Contributions:** C.L. derived the table in Appendix A and the mathematical proof validating the tensor notation. All the remaining parts are provided by D.M.

**Funding:** This research received no external funding.

**Acknowledgments:** The authors acknowledge Ergun Akleman for pointing out the Coons surface.

**Conflicts of Interest:** The authors declare no conflict of interest.

## Abbreviations

The following abbreviation is used in this manuscript:

ToC    Theory of Connections
PDE    Partial Differential Equations
ODE    Ordinary Differential Equations
IVP    Initial Value Problems
BVP    Boundary Value Problems

## Appendix A. All combinations of Dirichlet and Neumann constraints

| $c_{x,0}$ | $c_{0,y}$ | $c_{x,1}$ | $c_{1,y}$ | $c_{0,y}^x$ | $c_{1,y}^x$ | $c_{x,0}^y$ | $c_{x,1}^y$ | $v(x)$ | $v(y)$ |
|---|---|---|---|---|---|---|---|---|---|
| ✓ | ✓ | | | | | | | $\begin{Bmatrix} 1 \\ 1 \end{Bmatrix}$ | $\begin{Bmatrix} 1 \\ 1 \end{Bmatrix}$ |
| | ✓ | | | | | ✓ | | $\begin{Bmatrix} 1 \\ 1 \end{Bmatrix}$ | $\begin{Bmatrix} 1 \\ y \end{Bmatrix}$ |
| | | | | ✓ | | ✓ | | $\begin{Bmatrix} 1 \\ x \end{Bmatrix}$ | $\begin{Bmatrix} 1 \\ y \end{Bmatrix}$ |
| ✓ | ✓ | ✓ | | | | | | $\begin{Bmatrix} 1 \\ 1 \end{Bmatrix}$ | $\begin{Bmatrix} 1 \\ 1-y^2 \\ y^2 \end{Bmatrix}$ |
| ✓ | ✓ | | | | | | ✓ | $\begin{Bmatrix} 1 \\ 1 \end{Bmatrix}$ | $\begin{Bmatrix} 1 \\ 1 \\ y \end{Bmatrix}$ |
| ✓ | | ✓ | | ✓ | | | | $\begin{Bmatrix} 1 \\ x \end{Bmatrix}$ | $\begin{Bmatrix} 1 \\ 1-y \\ y \end{Bmatrix}$ |
| | | ✓ | | ✓ | | ✓ | | $\begin{Bmatrix} 1 \\ x \end{Bmatrix}$ | $\begin{Bmatrix} 1 \\ y-y^2 \\ y^2 \end{Bmatrix}$ |
| | ✓ | | | | | ✓ | ✓ | $\begin{Bmatrix} 1 \\ 1 \end{Bmatrix}$ | $\begin{Bmatrix} 1 \\ y-y^2/2 \\ y^2/2 \end{Bmatrix}$ |

| $c_{x,0}$ | $c_{0,y}$ | $c_{x,1}$ | $c_{1,y}$ | $c^x_{0,y}$ | $c^x_{1,y}$ | $c^y_{x,0}$ | $c^y_{x,1}$ | $v(x)$ | $v(y)$ |
|---|---|---|---|---|---|---|---|---|---|
| | | | | ✓ | | ✓ | ✓ | $\left\{\begin{array}{c}1\\x\end{array}\right\}$ | $\left\{\begin{array}{c}1\\y-y^2/2\\y^2/2\end{array}\right\}$ |
| ✓ | ✓ | ✓ | ✓ | | | | | $\left\{\begin{array}{c}1\\1-x\\x\end{array}\right\}$ | $\left\{\begin{array}{c}1\\1-y\\y\end{array}\right\}$ |
| | ✓ | ✓ | ✓ | | | ✓ | | $\left\{\begin{array}{c}1\\1-x\\x\end{array}\right\}$ | $\left\{\begin{array}{c}1\\y-y^2\\y^2\end{array}\right\}$ |
| | ✓ | | ✓ | | | ✓ | ✓ | $\left\{\begin{array}{c}1\\1-x\\x\end{array}\right\}$ | $\left\{\begin{array}{c}1\\y-y^2/2\\y^2/2\end{array}\right\}$ |
| | | ✓ | ✓ | ✓ | | ✓ | | $\left\{\begin{array}{c}1\\x-x^2\\x^2\end{array}\right\}$ | $\left\{\begin{array}{c}1\\y-y^2\\y^2\end{array}\right\}$ |
| | | | ✓ | ✓ | | ✓ | ✓ | $\left\{\begin{array}{c}1\\x-x^2\\x^2\end{array}\right\}$ | $\left\{\begin{array}{c}1\\y-y^2/2\\y^2/2\end{array}\right\}$ |
| | | | | ✓ | ✓ | ✓ | ✓ | $\left\{\begin{array}{c}1\\x-x^2/2\\x^2/2\end{array}\right\}$ | $\left\{\begin{array}{c}1\\y-y^2/2\\y^2/2\end{array}\right\}$ |
| ✓ | ✓ | | | | | ✓ | | $\left\{\begin{array}{c}1\\1\end{array}\right\}$ | $\left\{\begin{array}{c}1\\1\\y\end{array}\right\}$ |
| ✓ | | | | ✓ | | ✓ | | $\left\{\begin{array}{c}1\\x\end{array}\right\}$ | $\left\{\begin{array}{c}1\\1\\y\end{array}\right\}$ |
| ✓ | ✓ | | ✓ | | | ✓ | | $\left\{\begin{array}{c}1\\1-x\\x\end{array}\right\}$ | $\left\{\begin{array}{c}1\\1\\y\end{array}\right\}$ |
| ✓ | ✓ | ✓ | | | | ✓ | | $\left\{\begin{array}{c}1\\1\end{array}\right\}$ | $\left\{\begin{array}{c}1\\1-y^2\\y-y^2\\y^2\end{array}\right\}$ |
| ✓ | | ✓ | | ✓ | | ✓ | | $\left\{\begin{array}{c}1\\x\end{array}\right\}$ | $\left\{\begin{array}{c}1\\1-y^2\\y-y^2\\y^2\end{array}\right\}$ |
| ✓ | ✓ | | | | | ✓ | ✓ | $\left\{\begin{array}{c}1\\1\end{array}\right\}$ | $\left\{\begin{array}{c}1\\1\\y-y^2/2\\y^2/2\end{array}\right\}$ |

| $c_{x,0}$ | $c_{0,y}$ | $c_{x,1}$ | $c_{1,y}$ | $c^x_{0,y}$ | $c^x_{1,y}$ | $c^y_{x,0}$ | $c^y_{x,1}$ | $v(x)$ | $v(y)$ |
|---|---|---|---|---|---|---|---|---|---|
| ✓ | | | | ✓ | | ✓ | ✓ | $\left\{\begin{matrix}1\\x\end{matrix}\right\}$ | $\left\{\begin{matrix}1\\1\\y-y^2/2\\y^2/2\end{matrix}\right\}$ |
| ✓ | ✓ | | | | ✓ | ✓ | | $\left\{\begin{matrix}1\\1\\x\end{matrix}\right\}$ | $\left\{\begin{matrix}1\\1\\y\end{matrix}\right\}$ |
| ✓ | | | | ✓ | ✓ | ✓ | | $\left\{\begin{matrix}1\\x-x^2/2\\x^2/2\end{matrix}\right\}$ | $\left\{\begin{matrix}1\\1\\y\end{matrix}\right\}$ |
| ✓ | ✓ | | | ✓ | | ✓ | | $\left\{\begin{matrix}1\\1\\x\end{matrix}\right\}$ | $\left\{\begin{matrix}1\\1\\y\end{matrix}\right\}$ |
| ✓ | ✓ | ✓ | | ✓ | | ✓ | | $\left\{\begin{matrix}1\\1\\x\end{matrix}\right\}$ | $\left\{\begin{matrix}1\\1-y^2\\y-y^2\\y^2\end{matrix}\right\}$ |
| ✓ | ✓ | | | ✓ | | ✓ | ✓ | $\left\{\begin{matrix}1\\1\\x\end{matrix}\right\}$ | $\left\{\begin{matrix}1\\1\\y-y^2/2\\y^2/2\end{matrix}\right\}$ |
| ✓ | ✓ | ✓ | ✓ | | | ✓ | | $\left\{\begin{matrix}1\\1-x\\x\end{matrix}\right\}$ | $\left\{\begin{matrix}1\\1-y^2\\y-y^2\\y^2\end{matrix}\right\}$ |
| ✓ | | ✓ | ✓ | ✓ | | ✓ | | $\left\{\begin{matrix}1\\x-x^2\\x^2\end{matrix}\right\}$ | $\left\{\begin{matrix}1\\1-y^2\\y-y^2\\y^2\end{matrix}\right\}$ |
| ✓ | ✓ | | ✓ | | | ✓ | ✓ | $\left\{\begin{matrix}1\\1-x\\x\end{matrix}\right\}$ | $\left\{\begin{matrix}1\\1\\y-y^2/2\\y^2/2\end{matrix}\right\}$ |
| ✓ | | ✓ | | ✓ | | ✓ | ✓ | $\left\{\begin{matrix}1\\x-x^2\\x^2\end{matrix}\right\}$ | $\left\{\begin{matrix}1\\1\\y-y^2/2\\y^2/2\end{matrix}\right\}$ |
| ✓ | | | | ✓ | ✓ | ✓ | ✓ | $\left\{\begin{matrix}1\\x-x^2/2\\x^2/2\end{matrix}\right\}$ | $\left\{\begin{matrix}1\\1\\y-y^2/2\\y^2/2\end{matrix}\right\}$ |

| $c_{x,0}$ | $c_{0,y}$ | $c_{x,1}$ | $c_{1,y}$ | $c^x_{0,y}$ | $c^x_{1,y}$ | $c^y_{x,0}$ | $c^y_{x,1}$ | $v(x)$ | $v(y)$ |
|:---:|:---:|:---:|:---:|:---:|:---:|:---:|:---:|:---:|:---:|
| ✓ | ✓ | ✓ | ✓ | ✓ |  | ✓ |  | $\left\{\begin{array}{c}1\\1-x^2\\x-x^2\\x^2\end{array}\right\}$ | $\left\{\begin{array}{c}1\\1-y^2\\y-y^2\\y^2\end{array}\right\}$ |
| ✓ | ✓ |  | ✓ | ✓ |  | ✓ | ✓ | $\left\{\begin{array}{c}1\\1-x^2\\x-x^2\\x^2\end{array}\right\}$ | $\left\{\begin{array}{c}1\\1\\y-y^2/2\\y^2/2\end{array}\right\}$ |
| ✓ | ✓ |  |  | ✓ | ✓ | ✓ | ✓ | $\left\{\begin{array}{c}1\\1\\x-x^2/2\\x^2/2\end{array}\right\}$ | $\left\{\begin{array}{c}1\\1\\y-y^2/2\\y^2/2\end{array}\right\}$ |
| ✓ | ✓ | ✓ | ✓ |  |  | ✓ | ✓ | $\left\{\begin{array}{c}1\\1-x\\x\end{array}\right\}$ | $\left\{\begin{array}{c}1\\1-3y^2+2y^3\\y-2y^2+y^3\\3y^2-2y^3\\-y^2+y^3\end{array}\right\}$ |
| ✓ |  | ✓ | ✓ | ✓ |  | ✓ | ✓ | $\left\{\begin{array}{c}1\\-1+x\\1\end{array}\right\}$ | $\left\{\begin{array}{c}1\\1-3y^2+2y^3\\y-2y^2+y^3\\3y^2-2y^3\\-y^2+y^3\end{array}\right\}$ |
| ✓ |  | ✓ | ✓ | ✓ | ✓ | ✓ | ✓ | $\left\{\begin{array}{c}1\\x-x^2/2\\x^2/2\end{array}\right\}$ | $\left\{\begin{array}{c}1\\1-3y^2+2y^3\\y-2y^2+y^3\\3y^2-2y^3\\-y^2+y^3\end{array}\right\}$ |
| ✓ | ✓ | ✓ |  | ✓ |  | ✓ | ✓ | $\left\{\begin{array}{c}1\\1\\x\end{array}\right\}$ | $\left\{\begin{array}{c}1\\1-3y^2+2y^3\\y-2y^2+y^3\\3y^2-2y^3\\-y^2+y^3\end{array}\right\}$ |
| ✓ | ✓ | ✓ | ✓ | ✓ |  | ✓ | ✓ | $\left\{\begin{array}{c}1\\1-x^2\\x-x^2\\x^2\end{array}\right\}$ | $\left\{\begin{array}{c}1\\1-3y^2+2y^3\\y-2y^2+y^3\\3y^2-2y^3\\-y^2+y^3\end{array}\right\}$ |
| ✓ | ✓ | ✓ |  | ✓ | ✓ | ✓ | ✓ | $\left\{\begin{array}{c}1\\1\\x-x^2/2\\x^2/2\end{array}\right\}$ | $\left\{\begin{array}{c}1\\1-3y^2+2y^3\\y-2y^2+y^3\\3y^2-2y^3\\-y^2+y^3\end{array}\right\}$ |

| $c_{x,0}$ | $c_{0,y}$ | $c_{x,1}$ | $c_{1,y}$ | $c_{0,y}^{x}$ | $c_{1,y}^{x}$ | $c_{x,0}^{y}$ | $c_{x,1}^{y}$ | $v(x)$ | $v(y)$ |
|:---:|:---:|:---:|:---:|:---:|:---:|:---:|:---:|:---:|:---:|
| ✓ | ✓ | ✓ | ✓ | ✓ | ✓ | ✓ | ✓ | $\begin{Bmatrix} 1 \\ 1 - 3x^2 + 2x^3 \\ x - 2x^2 + x^3 \\ 3x^2 - 2x^3 \\ -x^2 + x^3 \end{Bmatrix}$ | $\begin{Bmatrix} 1 \\ 1 - 3y^2 + 2y^3 \\ y - 2y^2 + y^3 \\ 3y^2 - 2y^3 \\ -y^2 + y^3 \end{Bmatrix}$ |

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
