# Peer review of "The Multivariate Theory of Connections†"

_mathematics, doi:10.3390/math7030296_

Round 1

Reviewer 1 Report

The manuscript is devoted to the general problem of developing new mathematical optimisation instruments. Mortari's Theory of Connection is known enough from his other publications and presentations, as well as is both simple and effective for implementation. The paper includes development of analytical constrained expressions, as part of the Theory, aimed at transformation of  complicated two-dimensional constrained problems into ordinary unconstrained problems, which could be solved using existing mathematical methods. To add a little bit more content (or references) about implementation of this instrument for practical issues (in Astrodynamics, or other areas) is the only and not obligatory recommendation for this nice paper.  

Reviewer 2 Report

- Don't use references by numbers in brackets in the Absract. If its absolutely necessary to mention them use the author's surname and year of publication

- Add at the end of the Introduction a description of the organization of the manuscript section by section.

Round 2

Reviewer 2 Report

The manuscript in its present form is OK. I recommend publication